# Weed Role for Pollinator in the Agroecosystem: Plant–Insect Interactions and Agronomic Strategies for Biodiversity Conservation

**DOI:** 10.3390/plants13162249

**Published:** 2024-08-13

**Authors:** Stefano Benvenuti

**Affiliations:** Department of Agriculture, Food and Environment, University of Pisa, Via del Borghetto, 80, 56124 Pisa, Italy; stefano.benvenuti@unipi.it

**Keywords:** biological conservation, functional biodiversity, weed management, wildflowers, sustainability

## Abstract

The growing interest in safeguarding agroecosystem biodiversity has led to interest in studying ecological interactions among the various organisms present within the agroecosystem. Indeed, mutualisms between weeds and pollinators are of crucial importance as they influence the respective survival dynamics. In this review, the mutualistic role of flower visitors and the possible (often predominant) abiotic alternatives to insect pollination (self- and wind-pollination) are investigated. Mutualistic relations are discussed in terms of reward (pollen and/or nectar) and attractiveness (color, shape, scent, nectar quality and quantity), analyzing whether and to what extent typical weeds are linked to pollinators by rigid (specialization) or flexible (generalization) mutualistic relations. The entomofauna involved is composed mainly of solitary and social bees, bumblebees, Diptera, and Lepidoptera. While some of these pollinators are polylectic, others are oligolectic, depending on the shape of their mouthparts, which can be suited to explore the flower corollas as function of their depths. Consequently, the persistence dynamics of weed species show more successful survival in plants that are basically (occasional insect pollination) or totally (self and/or wind pollination) unspecialized in mutualistic relations. However, even weed species with typical abiotic pollination are at times visited during periods such as late summer, in which plants with more abundant rewards are insufficiently present or completely absent. Many typically insect-pollinated weeds can represent a valid indicator of the ecological sustainability of crop management techniques, as their survival dynamics are closely dependent on the biodiversity of the surrounding entomofauna. In particular, the presence of plant communities of species pollinated above all by butterflies (e.g., several Caryophyllaceae) gives evidence to the ecological compatibility of the previous agronomic management, in the sense that butterflies require certain weed species for oviposition and subsequent larva rearing and, therefore, provide further evidence of plant biodiversity in the environment.

## 1. Introduction

Weeds are predominantly self-pollinated [1]; insect-pollinated weeds are also frequently found in agricultural ecosystems [2]. Self-pollination is of crucial importance for rapid seed formation; in accordance with the time-limitation hypothesis [3], weeds require a certain degree of allogamy to maintain a genetic base capable of adapting to the dynamics of agronomic disturbance. Cross-pollination of angiosperms evolved in ancient natural ecosystems, from entomophily to anemophily [4], probably to reduce dependence on biotic factors whose presence is affected by erratic climatic conditions [5]. This widely accepted hypothesis is supported by the evidence of rudimental and inefficient that nectaries are often present even in typically wind-pollinated species [6]. However, despite this evolutionary trend, many agroecosystem plants base their survival dynamics on insect pollination. This mutualistic component of a part of the agroecosystem weed communities assumes a crucial ecological importance in terms of pollinator biodiversity even in this highly anthropized environment. The insects involved are often defined as “flower visitors” rather than pollinators, as their ecological role has not yet been fully clarified. It is frequently unclear whether the insect activity on the flowers allowing a contact pollen-gynoecium is accidental or mutualistic. However, the high frequency of flower visits constitutes a valid parameter for the almost certainty of this plant–insect mutualism [7]. Thus, for many insects and pollinated weeds, basically wildflowers, an evolution strategy towards flowers attractiveness can be discerned [8]. However, there are numerous cases of predominantly self- or wind-pollinated weeds that are visited by insects virtually capable of bringing about mutualistic or accidental gamy [9]. This ambophily is regarded as a transitional state intermediate between biotic and/or abiotic pollen movement [10]. This dual strategy allows the risk that erratic biotic and/or abiotic conditions may reduce the gene flow essential to evolve biotypes suitable for surviving in the agroecosystem. Such a strategy allows a shift from predominant self-pollination to predominant cross-pollination, as observed in numerous species [11]. A good example is offered by *Solanum ptycanthum* (Solanaceae), whose pollination strategy depends on the extent of disturbance in its growth environment, with predominantly insect-pollinated biotypes in natural ecosystems and predominantly self-pollinated biotypes in agricultural ecosystems [12]. It is not clear whether in this and other similar cases the greater attractiveness of the wild biotype is due to greater or lesser development and functioning of the nectaries. But it is worth noting that the presence of nectaries is not strictly necessary to induce insect flower visits, since pollen also constitutes a food source for a vast range of insects. Scanty or absent nectar production, thus, does not rule out the possibility that a species may be insect-pollinated, as it is pollen grain size that makes a species suited to insect pollination [13]. But it has been found that self-compatibility and self-pollination are associated with reduced pollen limitation, presumably because the capacity for self-fertilization decreases reliance on cross-pollination by pollinators [14].

Recent years have seen increasing interest in gene flow mechanisms (biotic and/or abiotic) between the various weed species, partly for agronomic reasons, such as herbicide resistance [15], and due to environmental concerns, as in the case of pollen transfer between genetically modified crops and potentially hybridizable weeds [16]. In addition, the plant and flower visitor interaction arising from insect pollination has aroused concern on account of increased awareness of the concept of safeguarding biodiversity in the agroecosystem [17] and in other anthropized ecosystems [18]. Attention is focusing on a possible cause–effect relationship between the rarefaction and/or disappearance of some species and their dependence on entomofauna, whose ecological role is often overlooked.

The purpose of this study was to survey the state of the art of flora–fauna interaction in weed communities in terms of insect pollination, investigate the ecological importance of these mechanisms in survival dynamics, and determine agronomic strategies that can be adopted to safeguard biodiversity in the agricultural environment.

## 2. Pollinator Biodiversity and Reward

Most flower visitors are social and solitary bees, bumblebees, Diptera, and Lepidoptera, as shown in Table 1. Each insect species feeds on pollen and/or nectar of given plant species as a function of its respective mouthparts [19], which, in many cases, have evolved in a manner that enables the insect to reach and feed on solid (pollen) or liquid (nectar) food. These food resources are produced in specific structures that are highly diversified among the various plant species [20]. 

Among flower visitors, honeybees (*Apis mellifera*, Figure 1) and solitary bees (Figure 2) are predominant. Within the Mediterranean environment, solitary bees are represented above all by Andrenidae, Anthophoridae, Apidae, Melittidae, Colletidae, Halictidae, and Megachilidae [35]. The reward consists of nectar and/or pollen, with the latter being transported in different body places and after being packed in special pollen baskets situated on the insects’ legs. But it is rare for bees to collect both pollen and nectar simultaneously, as energy economy prompts bees to visit species with a predominance of one or the other reward [36]. Natural ecosystems have an abundance of wildflowers, which have typically evolved nectar production as a reward. On the contrary, the conventional agroecosystem has a predominance of species poor of nectar, so that insects are rewarded with pollen. For example, *Papaver rhoeas*, in spite of their flower appearance, has no nectaries, and pollen is the only food source for pollinators, the latter being indispensable for seed set as this species is completely self-incompatible [37]. 

However, although species with brightly colored flowers are the most frequently visited, even many common weeds with less gaudy flowers also constitute a useful food source, especially during periods when the natural environment offers fewer species in flower [38]. Large numbers of species flower in spring [39], and they do compete with one another in producing nectar [40], while in the subsequent months, late-flowering species with less noticeable flowers may be of interest to the pollinator, even though the reward is less advantageous. Above all, in late summer, flower visitors may be observed on species that previously exerted poor visual attraction, as occurs in many Asteraceae, such as *Senecio vulgaris*, *Sonchus* spp., *Aster squamatus*, *Conyza canadensis*, and even *Xanthium strumarium*. The latter species, which is typically wind- and/or self-pollinated, constitutes a curious case in that the ecological role of flower visitors on pollination can be considered negligible, as the position of its male flowers, separate from the female flowers, suggests that pollen movement towards the gynaeceum may be purely accidental. The presence of flower visitors (in particular bees and bumblebees) has also been noted almost exclusively in late summer on other species with small and/or poorly attractive flowers, such as *Polygonum laphatipholium*, *P. aviculare*, *Cuscuta campestris*, *Portulaca oleracea*, *Stellaria media*, and *Abutilon theophrasti*. Similar phenomena have been observed on species with gaudy but usually self-pollinated flowers, such as *Convolvolus arvensis* and *Calystegia saepium* [41] and *Datura stramonium* [42].

A less important, but still underestimated, role is played by Diptera (Figure 3) (overall Syrphidae, Tachinidae, Sarcophagidae, and Bombyliidae) [43]. In Bombyliidae, on the other hand, the mouthparts appear to have evolved to allow utilization of nectar by means of a long proboscis that can penetrate inside small flowers [44]. Visits by Bombyliidae have been noted, above all, on flowers that would be difficult for other insects to reach due to their small and elongated floral calyx, as in the case of Gentianaceae (*Centaurium erythraea* and *Blackstonia perfoliata*), Campanulaceae (*Legousia speculum veneris* and *Jasione montana*), Lamiaceae (*Lamium purpureum* and *L. amplexicaule*), Primulaceae (*Anagallis arvensis* and *A. foemina*), and other species.

Some insects, above all, Lepidoptera, have a long proboscis that enables them to visit flowers with an elongated calyx even when the nectaries are hidden at the base of the calyx (Figure 4). Such insects include Lycaenidae, Pieridae and Sphingidae, Papilionidae, Nymphalidae, and Satyrids (Figure 5). Visits are frequent on flower species whose flowers have a particular shape, such as Dipsaceae (*Knautia arvensis* and *Dipsacus fullonum*) and Caryophyllaceae (*Agrostemma githago*, *Lychnis flos-cuculi*, *Silene* spp., etc.). However, pollen transport by butterflies is decidedly less efficient in comparison to bees [45]. Species belonging to the order of Coleoptera (Figure 6) are even less efficient, as such species, similarly to those of the order of Thysanoptera, often lack pollen transport specialization and appear to act more as pollen predators rather than potential pollinators [46]. Similarly poor efficiency is seen in ants, even though they are flower visitors of many species [47], and this appears to be due to the low pollen germination after ant contact [48]. Their lack of hair and their very limited plant–plant movement on account of their inability to fly, at least in most species, suggests that these Hymenoptera are likely to be only occasional pollinators. Indeed, it was observed that ants may negatively affect plant fitness by reduced intensity of pollinator visits and that ants are repelled from flowers of many plant species (overall in tropical environments), although this repellence is clearly not ubiquitous [49].

Similar scarce pollination-efficiency is evidenced by some nectar-robbing species such as *Bombus occidentalis*, which visits *Linaria vulgaris* where it collects nectar by poking holes into the corolla without penetrating inside it [50], and are likely to be of equally negligible ecological impact, since they are cheaters rather than mutualists [51].

## 3. Generalization or Specialization? 

It is widely believed that common weeds owe their time and space persistence to their lack of specialization [52], except for some species that are increasingly rare in conventional agroecosystems [53]. The evolutionary trend from generalization to specialization noted in natural ecosystems [54] does not appear to be suited to the requirements of weeds in an agricultural environment, where “plasticity” (despecialization) seems to be more important [55]. Indeed, it implies a lower risk of pollinator lack because of the high degree of disturbance dynamics, as typically occurs in the agroecosystem. Therefore, the specialization of some plant species towards certain pollinators could explain why they are increasingly uncommon [56]. Pesticide application, which is typical of conventional agricultural systems, can interfere with the efficacious but fragile mechanisms that involve rigid flora–fauna mutualisms. 

Indeed, “conventional” agroecosystems are characterized by the dominance of self- and wind-pollination, while, on the contrary, insect pollination is more frequent in natural ecosystems as a function of the greater abundance and biodiversity of pollinators typically available in undisturbed environments. However, even predominantly insect-pollinated species have different degrees of specialization depending on the possible pollinators [57], as floral symmetry plays an important role in plant–pollinator systems [58]. Zygomorphic flowers, such as *Consolida regalis*, *Echium vulgaris*, *Lamium amplexicaule*, and *Stachys arvensis*, are visited mainly by specialized long-tongued bees (Melittidae, Megachilidae, Anthrophoridae, and Apidae) as a consequence of the particular position of the nectaries [59]. Actinomorphic flowers, on the other hand, are visited by a higher number of visitor species. For example, almost all of the Asteraceae species (e.g., *Centaurea cyanus*, *Chrysanthemum myconis*, and *Cirsium arvense*) showed a higher degree of unspecialized visitors, such as short-tongued bees (Colletidae, Andrenidae, and Halictidae) and flies (personal observation). Another example is that of *Raphanus raphanistrum* (Brassicaceae), which exhibits traits typical of generalized pollination, including radially symmetric flowers, exposed reproductive organs, and an upright flower [60]. 

A further type of specialization, with visits limited to a restricted botanical group, may represent a characteristic of the pollinators themselves, as observed in Italy for *Heriades truncorum* (Megachilidae), which is almost always seen on Asteraceae weeds [61]. But it cannot necessarily be assumed that the insects most frequently observed on flowers are the most efficient pollinators, since pollen transport is strongly dependent on the shape and hair of the insect but also on the speed of visits, with rapid speed proving to be less efficient. For example, the typical rapid visits by long-tongued bees may result in reduced pollen transport [62]. Furthermore, efficiency of pollination is also influenced by the electrostatic forces of pollen, which can assure adhesion to the pollinator even if the insect may lack hair [63]. 

Overall, however, specialization is indisputably linked to flowering phenology [64], which may or may not be compatible with the biological cycle of the pollinator. Thus, with cool season flowering species, potential pollinators are represented by insects that are capable of maintaining a certain degree of activity even at low temperatures, as in the case of early Amaryllidaceae pollinated by cold-tolerant Andrenidae [65]. The role of temperature as a limiting factor is confirmed by observations on various species of Campanulaceae, which show decreasing frequency of visits with increasing altitude [66]. In addition, the literature indicates that pollinators tend to favor peak or earlier flowering, whereas predispersal seed predators tend to favor off-peak or later flowering [67].

An interesting example of plant–pollinator mutualistic specialization is found in *Silene noctiflora*, a gynomonoecious annual whose individuals produce both hermaphroditic and pistillate flowers. It flowers only during the night and is pollinated exclusively by nocturnal moths [68]. A similar system, albeit less exclusive, is seen in *Silene alba* [69], which tends to open its flowers at the end of the day, thereby allowing pollination both by diurnal (bees, flies, and wasps) and nocturnal visitors (the latter being mostly Sphyngid and noctuid moths). This mixed system is typical of numerous other species belonging to the same family of Caryophyllaceae, as in the genera *Agrostemma*, *Saponaria*, *Dianthus*, and *Vaccaria* [21]. 

Flora–fauna specialization does not depend only on the shape and manner of opening of the flowers but also on nectar composition, in terms of sugars and amino acids, as well as the nectar secretion rate, which is measurable in the field with various techniques [70]. For example, butterflies are attracted by the flowers richest in amino acids, as their diet is based exclusively on nectar and must therefore allow sufficient protein synthesis. Solitary and social bees are more attracted by an elevated sugar content, as they also feed on pollen and therefore do not need an additional protein supply (Gardener and Gilman, 2002). It has also been hypothesized that specialist nectarivores can assimilate sucrose, whereas some opportunistic nectar feeders digest only the simple exoses [71]. A crucial role is also played by amino acid typology. The predominance of phenylalanine and/or gamma-aminobutyric acid tends to attract long-tongued bees and flies (overall Syrphidae), whereas asparagine and tryptophane are rather repellent to these insect species [72]. Investigations aiming to obtain experimental evidence of an ecological function of nectar composition have been conducted on a vast range of species [73], with results suggesting that some amino acids elicit different responses in insect receptors. However, amino acid concentration in nectar is not exclusively a function of the genotype. It can be influenced by agronomic management, as in the case of nitrogen fertilization, which has been shown to increase nectar amino acid concentration in *Agrostemma githago* [74]. Natural factors such as arbuscular mycorrhizal fungi can, likewise, increase flower visitor numbers (overall visits by Diptera and Hymenoptera) in *Centaurea cyanus* [75]. Finally, the ecological significance of the toxic nectar secreted, for example, by *Heliotropium europaeum*, *Cuscuta* spp., *Solanum nigrum*, and *Euphorbia* spp. is still poorly understood. It has been hypothesized that bees are more resistant to alkaloids than adult Lepidoptera, and that alkaloids in nectar encourage pollination by specialist rather “flower-inconstant” pollinators [76].

Pollinator attraction is linked to the mechanisms involved in recognition of appropriate flowers. Recognition is crucial in that it avoids confusion in pollen transfer, which must take place within the same species as far as possible [77]. Flower shape and color both play an important role in facilitating recognition. Color is perceived differently by the insect as compared to the human eye, and light reflectance at wavelengths invisible to humans (roughly 300–400 nm) is well perceived [78]. Many Brassicaceae reflect ultraviolet color in order to attract the attention of pollinators [79]. Some flower colors appear to be correlated with certain categories of pollinators, although this cannot always be generalized due to poor convergence of data from different environments [80]. Specialist bumblebees have been noted to show a preference for purple, and this example would appear to confirm the so-called “pollination syndrome” theory [81]. In some cases, recognition is facilitated by color patterns: thus, the “search images” system possessed by insects [82] can be aided by characteristic black spots at the base of the petal, functioning as a “nectar guide” [83], as observed, for example, in *Papaver rhoeas*. But bright and gaudy colors are not always an indispensable condition for attracting flower visitors. Some Euforbiaceae have pale green flowers that do not stand out within the surrounding vegetation, yet they are frequently visited, as in the case of *Euphorbia esula* [84]. Moreover, a further and often decisive mechanism for identification and recognition of flowers consists of production and emission of fragrant molecules composed essentially of terpenoids and benzenoids [85]. 

## 4. Weed Pollination Strategies

The main characteristic of arable weeds is their ability to persist despite the vast range of agronomic disturbances. One of the crucial strategies adopted by weeds is rapid seed set. Rapidity is maximized with self-pollination, as autogamous seed set is not dependent on the occurrence of any (biotic or abiotic) event [86]. Weeds are regarded as pioneer flora of early stages of secondary successions [87], and their frequent annual cycle is often correlated with self-pollination [88] according to the “time-limitation” hypothesis of crucial importance in the typically highly disturbed agroecosystem. For example, *Amaranthus retroflexus*, characterized by flowers devoid of attractiveness and by almost total self-pollination [89], represents the ideotype of weeds as it already produces mature seeds just a few weeks after emergence [90]. Species whose corolla is a marked attractant (in terms of size and/or color) may also be mainly self-pollinated, as in *Convolvulus arvensis* [41], *Stellaria media* [91], *Portulaca oleracea* [92], and *Anagallis arvensis* [93]. However, it is likely that this predominance of self-fertilization is found in biotypes present in agricultural environments, while outcrossing is more widespread in biotypes present in less disturbed environments. This means that the frequent disturbances of the agroecosystem probably favored coevolution towards predominantly self-pollinated biotypes. This is the case of *Datura stramonium*, which is pollinated in its original environments by hawkmoths (*Manduca quinquemaculata*), Halictids, and honeybees (*Apis mellifera*) [94], whereas the biotypes present in cultivated fields are almost totally autogamous [95]. This coevolution with agronomic disturbances has led to a decrease in nectary function and a corresponding elongation of the androecium and gynoecium to allow contact between anthers and stigma (herkogamy), thereby favoring self-pollination [96]. Despite this, some exceptions that evolved towards self-incompatibility are observed, as in the case of *Papaver rhoeas* [97] and *Ranunculus repens* [98]. But, in general, the absolute self-incompatibility in the agroecosystem plant communities is unusual. Indeed, a variable frequency of distribution between self- and cross-pollination is more frequently found, as in the cases of *Anthemis cotula* [99], *Sinapis arvensis*, [100], *Raphanus raphanistrum* [101], and other insect-pollinated species typically widespread in the agroecosystem [102]. In the case of *Raphanus raphanistrum*, flower color (typically white or yellow) plays an ecological role in favoring or discouraging cross-pollination despite partial self-compatibility. Thus, one important visitor is a lepidopteran (*Pieris rapae*), which mainly visits yellow flowers, resulting in a predominance of cross-pollination in these populations [103], whereas self-pollination is predominant in white flower varieties. This dual typology of biotypes could represent an example of optimal trade-off between the advantages of one or the other gene flow mechanism, highlighting plastic and evolutionary changes in floral traits. Such a trade-off is not exclusive to this weed species but is widespread in many other species as well, and can probably be interpreted as a strategy for maintaining populations with diversified biological characteristics [104]. 

Overall, pollen self-incompatibility is one of the various strategies adopted by plant species to avoid the pollination of different flowers growing on the same plant, especially in the case of individuals bearing numerous blooms [105]. Such pollination, which would be pointless in terms of gene flow, is known as geitonogamy [106]. Other geitonogamy avoidance strategies include spatial and/or temporal separation of pollen and stigma. In general, geitonogamy avoidance is advantageous in favoring adaptability to dynamic environmental conditions [107]. A trade-off between geitonogamy and xenogamy (cross-pollination) is observed in *Daucus carota*, an andromonoecious and protandrous species [108]. In this case, separation of the male and female phases is complete at the level of the flower and umbel, but the two phases overlap at the level of the full plant, creating conditions for geitonogamy even if insect visits between the umbels of adjacent plants lead to xenogamy. Another curious characteristic of this species is the possible ecological role of the dark central floret of the inflorescences, for which a possible “fly catcher” role has been suggested, although its function has not yet been fully clarified [108]. Additional examples of a balance between self-pollination and insect pollination are observed in *Cynoglossum officinale*, *Echium vulgare* [109], and some species belonging to the botanical genus *Delphinum* [110], in which the extremely variable number of flowers results in diversified probability of geitonogamy. Plants with a greater number of flowers tend to favor self-pollination as there is an increased probability that the pollinators will sequentially visit (geitonogamy) flowers of the same plant. A further diversification of breeding frequency is found in *Echium vulgare*, as the protandrous flowers produce more nectar and receive higher rates of visitation during their male than during their female phase [111], although nectar production in this species is also closely dependent on environmental conditions [112].

Among geitonogamy avoidance strategies, a particularly drastic mechanism is displayed by dioecious species, in which the separation of individuals into different sexes makes self-fertilization impossible. An example is seen in *Silene dioica*, which is visited by bumblebees, hoverflies (Syrphidae), butterflies (mainly Pieridae), and honeybees [113]. But since the invasiveness of this species in the agroecosystem is negligible, as compared to other monoecious species belonging to the same genus, it can be concluded that such a strategy is unsuccessful [114].

## 5. Insect-Pollinated Weeds as Indicators of Biodiversity and Agroecosystem Health

A clear-cut distinction between entomopollinated and nonentomopollinated species cannot easily be drawn, as the true ecological role of insect visits has not yet been clarified for each individual species. This uncertainty is aggravated by the above-described differences among biotypes present in the wild (more dependent) versus those in the agroecosystem (less dependent). However, it is generally agreed that while the reward may consist of pollen (Figure 7), both quantitative and qualitative (sugar concentration) nectar production is linked to entomofauna through mutualistic specialization. Since nectar production requires considerable energy requirement, in short-lived weeds, characterized by annual cycle, nectar is less abundant than in perennial species [115], and if it is not collected, it is reabsorbed by the plant for its own metabolism [116]. It can, therefore, be stated that species with a well-developed nectary rely mainly on insect visits for their survival, with the visits being crucial for seed set. Thus, weed species characterized by this feature (essentially, wildflowers) face a greater risk within the agroecosystem, because the high level of disturbance of the agricultural environment tends to restrict the availability of their pollinators. Pesticide toxicity and its residues play a crucial role in this regard by severely affecting the chance for survival of entomofauna and, consequently, of insect-pollinated flora. 

The frequency of wildflower species in the agroecosystem plant communities can, therefore, represent a valid indicator of their ecological sustainability. It is now recognized that the floristic diversity of the agroecosystem provides an assessment of the agroecological impact [117]. This is particularly true about entomopollinated species, as their presence presupposes a level of biodiversity extended to the animal kingdom, and in this context, it should not be forgotten that evolution towards self-pollination occurred precisely in situations of a lack of pollinators [118].

The mutualistically more specialized species, such as wildflowers, constitute the most reliable ecological assessment parameters since these are the species whose presence is most severely threatened by disruption of the balance of the agroecosystem. Although the decreased presence or disappearance of some species may be due to other agronomic causes (herbicides, heightened aggression by more competitive weeds, crop seed selection, etc. [119]), it is highly probable that the declining numbers of pollinators have been a contributing factor in the increasing rarity of some species. As has been widely noted, in the past few decades, the decline of biodiversity has involved many previously widespread plant and animal organisms [120]. Throughout Europe, only plant species whose persistence dynamics do not rely on biotic action for pollination are only occasionally cited as rare weeds [121]. On the contrary, lists of declining species include numerous wildflowers, which are threatened by their dependence on flower visitors for seed set [122]. It has also been shown that the frequency of flower visitors on wildflowers is closely related to the quantity of viable seeds produced [123]. Weed species that are only scantily present have difficulty in attracting insects, as it has been noted both in natural ecosystems [124] and in the agroecosystem [125] that insects prefer to visit more numerous species. In arable fields, a large number of insect-pollinated wildflowers have now become rare or are in decline, such as *Agrostemma githago*, *Centaurea cyanus*, *Papaver argemone*, *Ranunculus arvensis*, [126], *Chrysanthemum sagetum*, *Matricaria recutita*, *Legousia hybrida*, *Silene alba*, *Viola arvensis* [127], *Consolida regalis*, *Silene noctiflora*, *Lamium amplexicaule*, [128], *Myosotis arvensis*, *Viola tricolor* [129], *Legousia speculum veneris*, *Anchusa arvensis*, [130], *Nigella arvensis*, and *Ornithogalum umbellatum* [131].

As stated above, the risk of decline is greatest when mutualistic interaction is specialized. Thus, many of the abovementioned species belong to the Cariophyllaceae, a botanical family often characterized by rigid mutualistic interactions set in motion by butterflies [132]. This type of mutualism is highly fragile, because Lepidoptera require a twofold plant-related availability: food source (visitable flora endowed with nectaries), and suitable conditions for reproduction (appropriate flora for oviposition and feeding of larval forms). Each butterfly species is dependent on restricted plant groups, often belonging to a single botanical family, a single genus, or even a single species (Table 2). Generally, reference is made to a hierarchy of preferences since some species may be preferred to others within a given botanical grouping, as in the case of *Papilio machaon*, which oviposits exclusively on Apiaceae (Figure 8). If certain host plants have poor invasiveness within the various ecosystems, this inevitably leads to very scanty presence not only of the respective plant species but also of the correlated butterfly species [133]. 

While specificity between pollinator and host weed is may be variable, it is quite typical of the different families of Lepidoptera, being found preferentially or obligatorily linked to restricted botanical groupings. For example, with Satytiridae (e.g., *Brintesia circe*), there is only scanty specialization as the pollinator/host-plant relation is observed in many ubiquitarian species of Graminaceae, but requirements are more stringent for *Macroglossa stellatarum* (Sphyngidae), which needs one of the various species of the genus *Galium* (Rubiaceae). The risk of butterfly/plant-host coextinction cannot be disregarded and has already been reported in some parts of the world [143]. It is also interesting to note that numerous Lepidoptera, known as myrmecophilous butterflies, have mutualistic relations with ants, as ants defend butterfly eggs and the subsequent caterpillars against predator attack [144]. It has been observed that oviposition of myrmecophilous butterflies takes place preferentially on plants most frequently visited by ants [145], showing that the presence of butterflies testifies to an even more extensive level of biodiversity [146]. In contrast, bumblebee reproduction, while similarly limited by scanty availability of undisturbed environments, is less specialized as it generally takes place in soil [147]. Lack of specificity is also noted in most solitary bees, with some species nesting in soil while others also nest in plant residue cavities [148].

A special form of ecological interaction is found in pollinators that require the presence of other pollinators for their reproduction. For example, in many species of Diptera Bombylidae, which have a very thin and elongated mouthpiece allowing them to collect nectar from small flowers such as *Anagallis arvensis*, *Legousia speculum veneris*, *Centaurium erytrea*, and *Lamium purpureum* (Benvenuti, personal observation), parasitic oviposition takes place in the nest of several species of solitary bees [149], where the larvae feed on resources intended to be food for the bees. Indeed, Diptera Bombylidae exhibit mutualistic behavior towards insect-pollinated plant species but have parasitic behavior towards other pollinators.

Often, rare weeds are pollinated above all by Lepidoptera, as in the case of many Caryophyllaceae [150]. Consequently, they represent the most valid indicator of the biodiversity of the agroecosystem since their presence testifies to a complex level of flora–fauna interactions. But exceptions to this rule are found for some butterflies, such as *Pieris rapae*, which often choose the same species both for pollinating and for oviposition as well as for subsequent rearing of larvae. This exemplifies mutualism and parasitism simultaneously [151], which may result in a sort of conflict of interest [152] for the pollinator itself. However, apart from these exceptions, butterfly presence and diversity depend on the landscape context in that their survival dynamics are linked to availability of the required host species in the environment [153]. Analogously, the widespread presence of Syrphid (Diptera) has also been considered a good indicator of plant biodiversity [154], on account of the abundance of different environments suitable for their reproduction. Some species of spider can also be considered as a further parameter for assessment of ecosystem integrity, as they hide on flowers to prey on pollinators. Such a phenomenon has been observed on *Aslepias syriaca* [155], *Leucanthemum vulgare* [156], and many other species. Thus, spiders represent the tip of the ecological pyramid of this flower–pollinator–predator food chain, and this ecosystem appears be particularly vulnerable [157].

Confirmation of the reliability of insect-pollinated flora as an indicator of ecological sustainability of the agronomic cultural practices adopted comes from the observation that organic agricultural systems lead to an increase in insect-pollinated species [158]. Future ecological assessments could be based on monitoring the spider species known to be the preferential predators of pollinating insects.

## 6. Long-Term Plant–Animal Biodiversity Sustainability

Weed management is crucial for biodiversity sustainability [159], as also highlighted by organic agricultural systems [160]. Perhaps the starting point is diversification of land management, since species richness, genetic variability, and extinction probability are closely linked to landscape traits such as habitat diversity, structural heterogeneity, patch dynamics, and perturbations [161]. In other words, arable weed diversity increases with landscape complexity [162]. Landscape planning is crucial for biodiversity [163]. It is important to keep in mind that distances between nesting environments and food sources must not be excessive, since beyond a certain distance, the trip energy will no longer be advantageous due to excessive energy consumption for flight between the nest and flora to be visited. Other, more suitable environments will therefore be sought [164]. This implies that the geometry of the agroecosystem plays an important role in ensuring that insect-pollinated weeds achieve sufficient seed set. 

Woods represent a fundamental reserve of environments suitable for pollinator survival, especially if they are established in a mosaic pattern [165] within an agricultural setting. A similar positive effect is produced by cattle grazing [166] because this kind of land use presupposes both forage resources and nesting resources [167]. With regard to the potential for nesting, at least as far as numerous species of solitary bees and bumblebees, it is important to plan uncropped areas near the crops in such a way as to allow undisturbed aboveground nesting [168]. In addition, the introduction of long-term crops such as Lucerne (*Medicago sativa*) can guarantee prolonged periods (3–4 years) free from soil disturbance. Honeybees are severely damaged by microencapsuled pesticides, whose microgranules adhere to the insect’s hair and are thereby transferred into the hives, where their toxic effects can kill the larvae [169]. Therefore, the use of such pesticides should be strictly avoided.

The likelihood of pollinator survival can be increased by the presence of field margins, hedges [170], and other buffer zones [171] or set-aside fields [172]. Such areas not only offer a suitable environment for soil-nesting pollinators, but also for Lepidoptera that require certain weeds on which to oviposit [173]. Wildflowers linked to mutualistic relations with the pollinators represent the ideotype of field margins as they not only provide a suitable ecological niche for an elevated number of pollinators, but they also ensure positive benefits for the agricultural landscape. Thus, it has been shown that the introduction of complex mixed wildflower strips leads to an increase in butterflies, which are drawn by the presence of host plants for oviposition and nectar as a food source [174]. The use of native wildflowers achieves the best ecological response in safeguarding pollinator biodiversity, above all, with regard to specialized pollinators (i.e., short- or long-tongued bees) linked to the wildflowers involved [175]. The introduction of exotic pollinators [176] tends to impair the plant–pollinator balance of the agroecosystem overall following increasingly evident climate changes [177]. This occurs because the balance of weed–pollinator competition is disrupted by altered mutualistic plant–animal interactions [178]. Even though it is not always easy to distinguish the cause from the effect, i.e., to determine whether the imbalance is triggered by a lack of wildflowers or pollinators [179], limiting or preventing the introduction of non-native animal or plant organisms into such environments represents a strategy of paramount importance. It may, therefore, help to avert the tendency to “biological globalization” and the ensuing genetic erosion or loss of native insect-pollinated plants and/or the relative flower visitors.

## 7. Conclusions 

Every weed species is characterized by a particular survival strategy and ruderal species (early flowering, abundant and prolonged seed production) derived from an evolutionary direction capable of giving them ideal weed traits [180] to persist in the agroecosystem. In this context, the rigid plant–pollinator mutualisms are undoubtedly a disadvantage since they presuppose the presence of a consistent pollinator quantity and biodiversity. Despite this, in many cases, there is a transition towards weed communities that display an increasing presence of species whose survival strategies depend on pollinators with varying degrees of specialization. The growing need to maintain and/or restore agroecosystem biodiversity has focused attention on the insect-pollinated weeds that are among the first to decline or even disappear in the agricultural landscape. Paradoxically, it is now widely believed that even weeds perform an “ecosystem service” dedicated to the survival of pollinators essential for the productivity of the various insect-pollinated crops [181]. Within pollinators, butterflies are particularly subjected to rarefaction since their survival does not depend exclusively on the presence of pollinated plants but also on the further availability of host plants essential for their oviposition. Thus, it is important to identify valid ecological indicators to monitor the health of the agroecosystem. The presence of crab spiders has been proposed as a valid indicator of the level of the agroecosystem biodiversity. Indeed, these arthropods feed on the pollinators that wait camouflaged on the flowers, thus highlighting the biodiversity of both pollinated plants and pollinators [182] (Figure 9). However, further studies are required to determine more precisely, for each species, the pollinator requirement for seed set, especially for specialized weeds. This improved knowledge would not only aid research based on biological indicators, but it would also help to optimize the biodiversity restoration programs of degraded agroecosystems.

## Figures and Tables

**Figure 1 plants-13-02249-f001:**
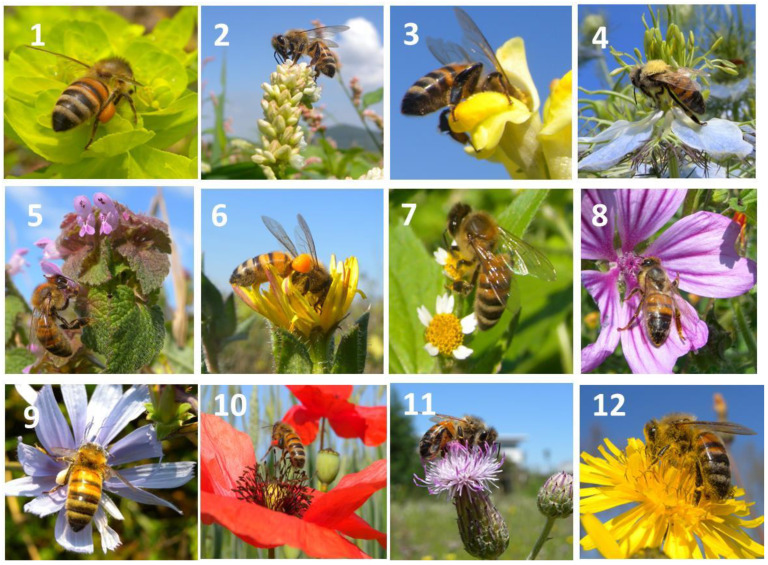
Honeybees observed on the flowers of common weeds of the agroecosystem: 1 = *Euphorbia helioscopia*, 2 = *Polygonum laphatifolium*, 3 = *Linaria vulgaris*, 4 = *Nigella damascena*, 5 = *Lamium purpureum*, 6 = *Picris echioides*, 7 = *Galinsoga parviflora*, 8 = *Malva sylvestris*, 9 = *Cichorium inthybus*, 10 = *Papaver rhoeas*, 11 = *Cirsium arvense*, 12 = *Crepis vesicaria*.

**Figure 2 plants-13-02249-f002:**
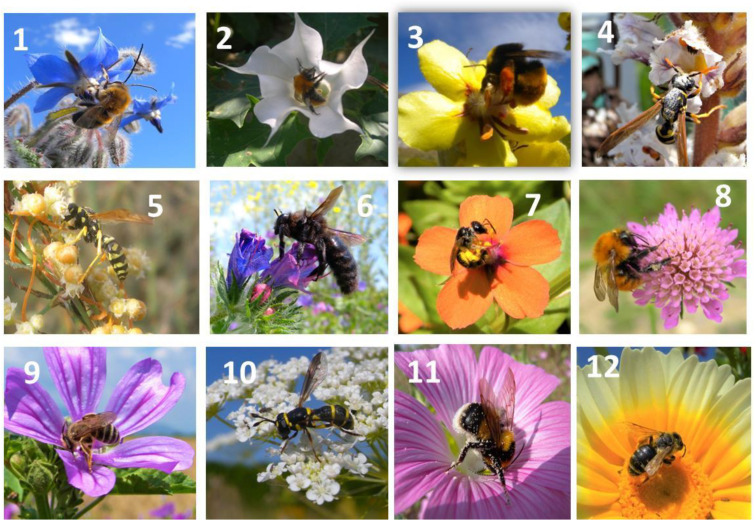
Bumblebees (2, 3, 8, 11), solitary bees (1, 6, 7, 9, 12), and wasps (4, 5, 10) observed on common weeds of the agroecosystem: 1 = *Borago officinalis*, 2 = *Datura stramonium*, 3 = *Verbascum sinuatum*, 4 = *Orobanche crenata*, 5 = *Cuscuta campestris*, 6 = *Echium vulgare*, 7 = *Anagallis arvensis*, 8 = *Scabiosa columbaria*, 9 = *Malva sylvestris*, 10 = *Daucus carota*, 11 = *Lavatera punctata*, 12 = *Chrysanthemum coronarium*.

**Figure 3 plants-13-02249-f003:**
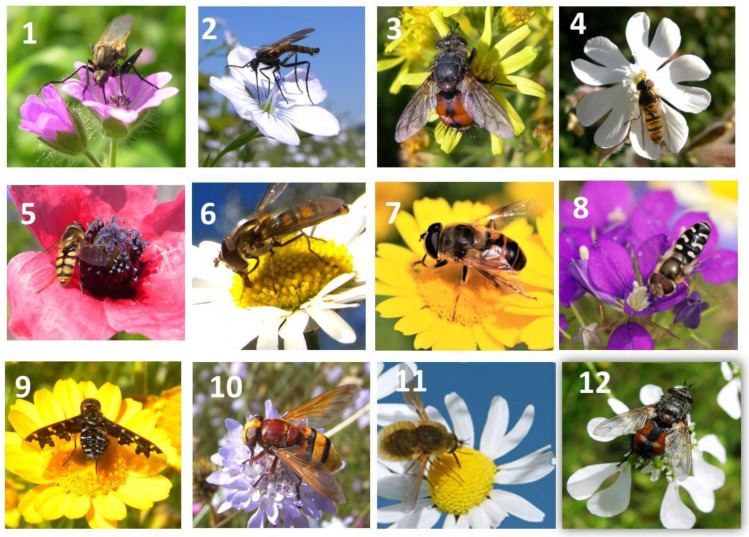
Diptera Empididae (1, 2), Tachinidae (3, 12), Syrphidae (4, 5, 6, 7, 8, 10), and Bombyiliidae (9, 11) observed on common weeds of the agroecosystem: 1 = *Geranium molle*, 2 = *Linum perenne*, 3 = *Inula viscosa*, 4 = *Silene alba*, 5 = *Papaver hybridum*, 6 = *Anthemis arvensis*, 7 = *Crysanthemum segetum*, 8 = *Scandix pecten-veneris*, 9 = *Cruysanthemum coronarium*, 10 = *Cephalaria transsylvanica*, 11 = *Matricaria chamomilla*, 12 = *Tordylium apulum*.

**Figure 4 plants-13-02249-f004:**
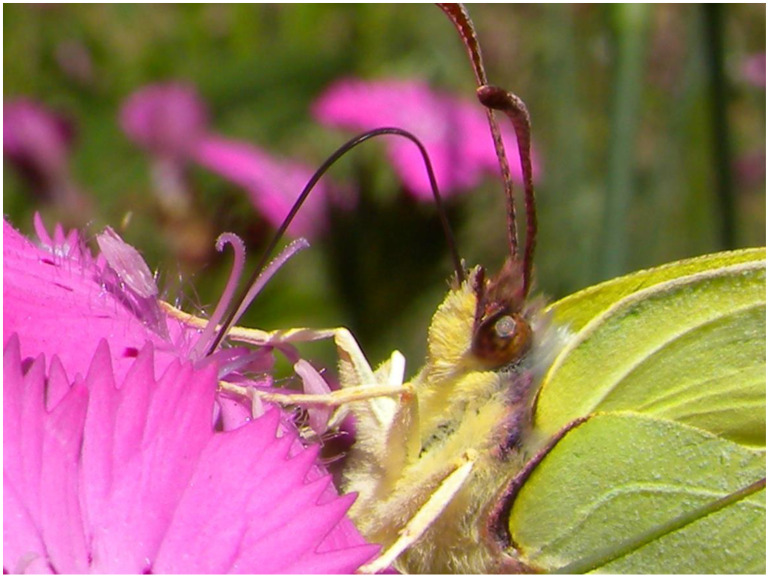
Butterfly (*Gonepteryx rhamni*) observed on *Dianthus cartusianorum* flower during nectar suction through their long-proboscid, evolved to be able to reach the nectaries of elongated floral calyxes.

**Figure 5 plants-13-02249-f005:**
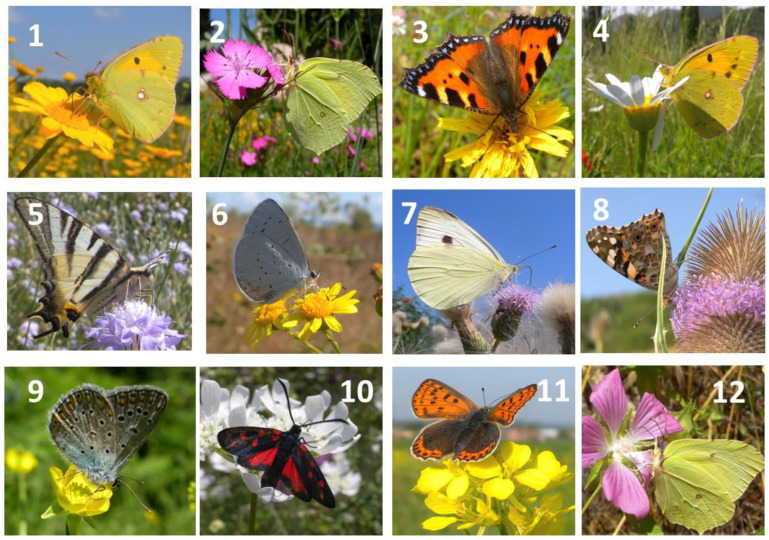
Lepidoptera observed on common wildflowers of the agroecosystem: 1 = *Crysanthemum segetum*, 2 = *Dianthus carthusianorum*, 3 = *Crepis vesicaria*, 4 = *Anthemis arvensis*, 5 = *Cephalaria transsylvanica*, 6 = *Senecio erraticus*, 7 = *Cirsium arvense*, 8 = *Dipsacus fullonum*, 9 = *Ranunculus arvensis*, 10 = *Orlaya grandiflora*, 11 = *Sinapis arvensis*, 12 = *Lavatera punctata*.

**Figure 6 plants-13-02249-f006:**
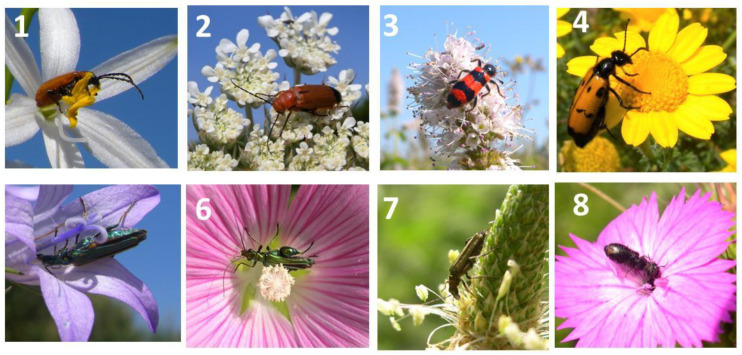
Coleoptera observed on common weeds of the agroecosystem: 1 = *Ornithogalum umbellatum*, 2 = *Ammi majus*, 3 = *Mentha suaveolens*, 4 = *Crysanthemum segetum*, 5 = *Campanula rapunculus*, 6 = *Lavatera punctata*, 7 = *Plantago lanceolata*, 8 = *Dianthus cartusianorum*, 9 = *Daucus carota*, 10 = *Malva sylvestris*, 11 = *Anagallis arvensi*, 12 = *Ornithogalum umbellatum*.

**Figure 7 plants-13-02249-f007:**
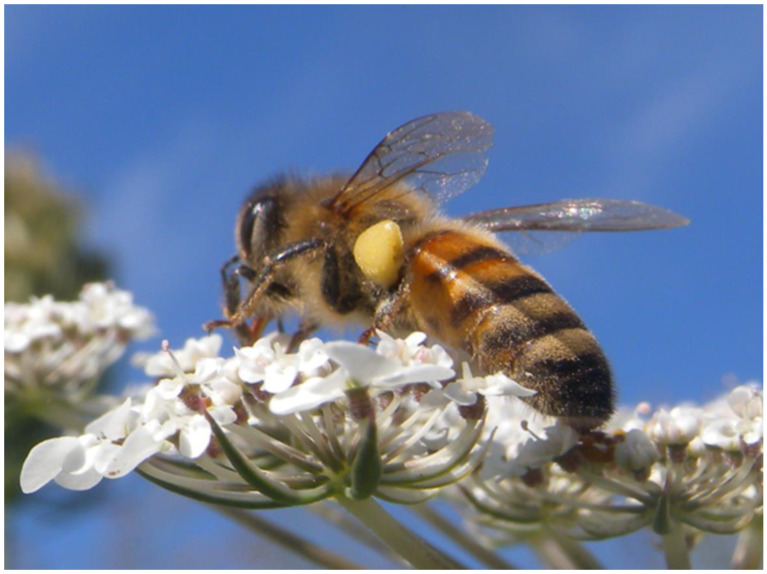
Honeybee observed during a visit to an inflorescence of *Daucus carota*: note the balls of pollen accumulated on the hind legs, typically yellow in this species.

**Figure 8 plants-13-02249-f008:**
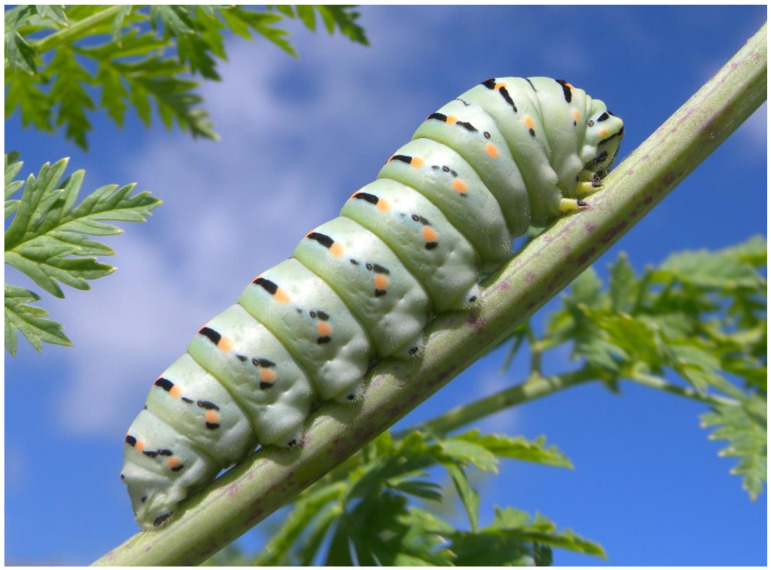
Caterpillar of the *Papilio machaon* butterfly specialized to lay eggs on plants of the Apiaceae family (in this case, the toxic *Conium maculatum*).

**Figure 9 plants-13-02249-f009:**
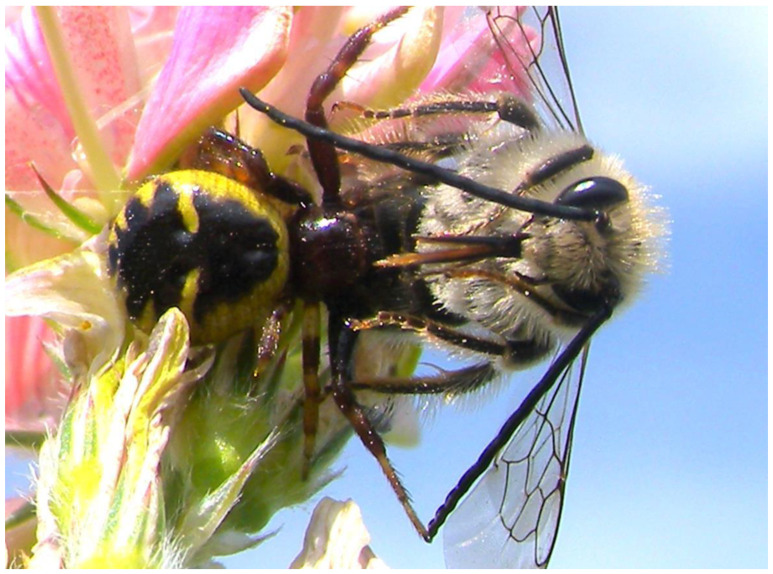
Observed crab spider with their evident success in catching a solitary bee on *Onobrychis viciifolia* inflorescence.

**Table 1 plants-13-02249-t001:** Flower visitor observations on several weed species of the agroecosystem.

Weed Species	Observed Flower Visitors	Source
*Abutilon theophrasti* Medic.	Solitary bees, Tachinidae	Benvenuti, personal observ.
*Agrostemma githago* L.	Lepidoptera, solitary bees	[21]
*Anagallis arvensis* L.	Solitary bees, Bombylidae, Coleoptera	Benvenuti, personal observ.
*Anthemis cotula* L.	Solitary bees, Lepidoptera	Benvenuti, personal observ.
*Arum italicum* Mill.	Diptera	[22]
*Asclepias syriaca* L.	*Bombus*, *Xylocopa*, Sphecidae	[23]
*Aster squamatus* (Spreng.) Hier.	Bees	Benvenuti, personal observ.
*Blackstonia perfoliata* (L.) Huds.	Bombylidae	Benvenuti, personal observ.
*Borago officinalis* L.	Bees, solitary bees	Benvenuti, personal observ.
*Calystegia sepium* (L.) R.Br.	Solitary bees, Coleptera	Benvenuti, personal observ.
*Centaurea cyanus* L.	Bees, solitary bees. Lepidoptera	[24]
*Centaurium erytrea* Rafn	Syrphidae	[25]
*Chrysanthemum segetum* L.	Bees, solitary bees; Bomyilidae; Lepidoptera	[26]
*Chrysanthemum coronarium* L.	Bees, solitary bees; Bomyilidae; Lepidoptera	Benvenuti, personal observ.
*Cirsium arvense* (L.) Scop.	Solitary bees, Lepidoptera	[27]
*Consolida regalis* Gray.	*Bombus*, Lepidoptera	[26]
*Convolvulus arvensis* L.	Solitary bees, Coleptera	Benvenuti, personal observ.
*Conyza canadensis* L.	Bees	Benvenuti, personal observ.
*Cuscuta campestris* Yunk.	Bees, wasps	Benvenuti, personal observ.
*Cychorium inthibus* L.	Diptera, solitary bees	Benvenuti, personal observ.
*Datura stramonium* L.	*Bombus* spp.	Benvenuti, personal observ.
*Dianthus carthusianorum* L.	Lepidoptera	[26]
*Dipsacus fullonum* L.	Lepidoptera, solitary bees	Benvenuti, personal observ.
*Echium vulgare* L.	*Bombus* spp.	[26]
*Euphorbia esula* L.	Solitary bees	[28]
*Euphorbia helioscopia* L.	Bees	Benvenuti, personal observ.
*Geranium molle* L.	Diptera	Benvenuti, personal observ.
*Lamium purpureum* L.	Bees, solitary bees, Bombylidae	Benvenuti, personal observ.
*Lavatera punctata* All.		[26]
*Legousia speculum-veneris* (L.) *Chaix*	Bees, Syrphidae, Bombylidae	[24]
*Linaria vulgaris* Mill.	* Bombus*	[29]
*Lycnis flos-cuculi* L.	Bees, solitary bees, Lepidoptera	[26]
*Matricaria chamomilla* L.	Bees, solitary bees, Bombylidae, Lepidoptera	Benvenuti, personal observ.
*Nigella damascena* L.	Bees	[26]
*Oenotera biennis* L.	Lepidoptera	[30]
*Ornithogalum* spp.	Solitary bees, Coleoptera	Benvenuti, personal observ.
*Papaver rhoeas* L.	Bees, solitary bees, *Xilocopa*	Benvenuti, personal observ.
*Polygonum lapathifolium* L.	Bees, solitary bees	Benvenuti, personal observ.
*Portulaca oleracea* L.	Bees, Bombylidae	Benvenuti, personal observ.
*Senecio vulgaris* L.	Bees	Benvenuti, personal observ.
*Silene alba* (Mill.) Krause	Nocturnal and diurnal Lepidoptera	[31]
*Silene dioica* (L.) Clairv.	Bombus, bees, Lepidoptera, Syrphidae	[32]
*Silene noctiflora* L.	Nocturnal Lepidoptera	[33]
*Sinapis arvensis* L.	Bees, solitary bees; Bomyilidae; Lepidoptera	Benvenuti, personal observ.
*Stellaria media* (L.) Vill.	Bees, solitary bees	Benvenuti, personal observ.
*Veronica persica* Poir.	Solitary bees, ants	Benvenuti, personal observ.
*Viola* spp.	Solitary bees	[34]
*Xanthium strumarium* L.	Bees	Benvenuti, personal observ.

**Table 2 plants-13-02249-t002:** Some examples of weed host selected by several butterfly species.

Weed-Host	Butterfly Species	Source
*Amaranthus* spp., *Chenopodium* spp.	* Pholisora catullus*	[134]
Apiaceae	* Papilio machaon*	[135]
*Aristolochia* spp.	* Pachliopta aristolochiae*	[136]
*Carex* spp.	* Lopinga achine*	[137]
*Asteraceae*, *Malvaceae*, others	* Vanessa cardui*	[134]
*Avena fatua* L.	* Cercyonis pegala*	[134]
*Bidens pilosa* L.	* Nathalis iole*	[134]
*Brassica nigra* (L.) W.D.J.Koch	* Pontia bekeri*	[134]
Brassicaceae	* Anthocaris sara*	[134]
Brassicaceae	* Euchloe ausoides*	[134]
Brassicaceae	* Pieris rapae*	[134]
Brassicaceae	* Pontia protodice*	[134]
*Cardamine* spp., *Nasturtium* spp.	* Anthocaris cardamines*	[138]
Caryophyllaceae	* Euphyia picata*	[139]
*Centaurea* spp.; *Plantago* spp.	* Melitaea didyma*	[139]
*Chenpopodium album* L.	* Brephidium exilis*	[134]
*Cirsium* spp., *Centaurea solstitialis*	* Phycioides mylitta*	[134]
*Cirsium* spp., *Centaurea* spp.	* Tymelicus sylvestris*	[139]
*Echium vulgare* L.	* Ethmia terminella*	[139]
*Galium* spp.	* Ryparia purpurata*	[139]
*Galium* spp.	* Macroglossa stellatarum*	[139]
Graminaceae	* Brintesia circe*	[139]
Graminaceae	* Lerodea eufala*	[134]
*Hypericum* spp.	* Deilephila elpenor*	[139]
Lamiaceae	* Perizoma alchemillata*	[139]
*Lytrum* spp., *Epilobium* spp.	* Hyles vespertilio*	[139]
* Malva sylvestris*, *Alcea rosea*	* Heliopetes ericetorum*	[134]
*Malvaceae*, *Chenopodium album* L.	* Pyrgus communis*	[134]
* Malvaceae*, *Papilionaceae*	* Strymon melinus*	[134]
* Malvaceae*, *Urticaceae*	*Vanessa annabella*,	[134]
*Melilotus officinalis* Lam., *M. alba* Medic.	*Colias eurytheme*	[134]
Papillionaceae	*Cyaniris semiargus*	[139]
Plantaginaceae	*Euphydryas chalcedona*	[140]
*Plantago lanceolata* L.	*Mellicta hatalia*	[141]
*Plantago major* L. and *P. lanceolata* L.	*Euphydryas editha*	[134]
Polygonaceae	*Lycaena phlaeas*	[139]
*Polygonum persicaria* L.	*Lycaena helloides*	[134]
*Rumex crispus* L.	*Lycaeana xanthoides*	[134]
*Rumex* spp.	*Lycaena cupreus*	[134]
*Scrophulariaceae*	*Junonia coenia*	[134]
*Scrophulariaceae*, *Verbenaceae*	* Junonia coenia*	[140]
*Silene* spp.; *Lycnis* spp.	* Hadena rivularis*	[139]
*Sonchus oleraceus* L.	* Helicoverpa armigera*	[142]
*Urtica* spp.	* Vanessa atlanta*	[139]
*Veronica* spp.	* Stenoptilia pterodactyla*	[139]
*Viola* spp.	* Argynnis paphia*	[139]

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
