# Peer review of "Weed Role for Pollinator in the Agroecosystem: Plant–Insect Interactions and Agronomic Strategies for Biodiversity Conservation"

_plants, 2024, doi:10.3390/plants13162249_

Round 1

Reviewer 1 Report

Comments and Suggestions for Authors

The present manuscript is a review of the Weeds as Pollinator host plants and their role in the agroecosystem.

It  is a valuable contribution to the understanding of weed-pollinator interactions and their role in biodiversity conservation within agroecosystems. It aligns well with existing literature and offers practical solutions for enhancing ecological sustainability. By emphasizing the mutualistic relationships between weeds and pollinators, the study advocates for a shift in weed management practices to support biodiversity and ecological health.

Overall, this manuscript is a significant and timely addition to the field of agroecology and biodiversity conservation, offering insights that are crucial for developing sustainable agricultural systems.

Nevertheless, it will benefit from a linguistic revision and a literature update. In the attached file we provide specific comments/suggestions/corrections, which we hope that will improve the manuscript.

Comments on the Quality of English Language

While the english are well used, in general, the manuscript will benefit from a linguistic revision. 

Author Response

Comments 1: The present manuscript is a review of the Weeds as Pollinator host plants and their role in the agroecosystem.
It  is a valuable contribution to the understanding of weed-pollinator interactions and their role in biodiversity conservation within agroecosystems. It aligns well with existing literature and offers practical solutions for enhancing ecological sustainability. By emphasizing the mutualistic relationships between weeds and pollinators, the study advocates for a shift in weed management practices to support biodiversity and ecological health.
Overall, this manuscript is a significant and timely addition to the field of agroecology and biodiversity conservation, offering insights that are crucial for developing sustainable agricultural systems.

Response 1: Thanks for the comments. I have inserted the instructions I received from the PDF that indicated the corrections to be made.

Comments 2: Nevertheless, it will benefit from a linguistic revision and a literature update. In the attached file we provide specific comments/suggestions/corrections, which we hope that will improve the manuscript.

Response 2: Ok, I simplified some sentences according to a synthetic language

Reviewer 2 Report

Comments and Suggestions for Authors

This article deals with the very important issue of pollination of plant flowers. The review article is of relevance to many plant pollinator researchers. Despite the strengths of the manuscript, I found significant weaknesses and points of discussion that the author should consider.

1. Too much of the text (17%) is intensively re-written from two sources, including citations. It is necessary to restructure these parts of the text to remove suspicions of possible plagiarism (see iTenticate analysis). 

2. Some terms are not used correctly in the text. One such term is 'flora'. For example (line 23) it says: 'in which flora with more abundant rewards'. Flora is a totality of species, an abstract, and therefore cannot be given the characteristics of plants.  The term 'weed' is also used too often, sometimes even when it is not needed. The neutral term 'plant' or 'species' can be used, without negative connotations. Moreover, there is the question of what the author means by 'weed'. Can Dianthus carthusianorum really be a 'weed' in the classical sense of the term? I think that the concept of the terms, if the author interprets them freely, should be discussed separately. 

3. The article is profusely illustrated with original (not credited, but I assume original) photographs. I question whether all the illustrations are really necessary, and whether they are so large (Fig. 4, Fig. 7, Fig. 8, Fig. 9). In addition, some of the plants in the photos are misidentified. For example, Fig. 1 (8) is not Malva officinalis (which is no longer the name of the species) but Malva sylvestris; Fig. 1 (11) is not Cirsium vulgare but Cirsium arvense; Fig. 2 (2), not Datura stramonium but Datura innoxia; Fig. 3, the entries are all confused. Other figures also contain misidentified plants. All figures need to be checked and their captions corrected.

4. It is not clear on which source the nomenclature of plants and insects is based. I believe that a Materials and Methods section should be drafted, which should indicate what the nomenclature is based on, how the sources for the analysis have been chosen and other methodological details. For example, about one third of the plant names given in Table 1 do not correspond to modern taxonomy (Anagallis arvensis,  Aster squamatus, Chrysanthemum segetum, Chrysanthemum coronarium, Conyza canadensis, etc.). The same applies to the captions of the illustrations and the names of the taxa used in the text. 

5.  I think that the use of vernacular plant names in the text is redundant. They do not provide any information and in some cases only cause confusion. We are well aware that English vernacular names vary widely from one English-speaking country to another, not to mention other countries where English is not the main language. In a scientific article, scientific plant names are quite sufficient.

6. Is it not possible to specify the species, or at least the genera, of the insects in the photographs? For example, Fig. 6 has many beetles but no names. 

7. Agroecosystems are composed not only of weeds but also of crops. I believe that the impact of crops on the protection of pollinator insect diversity should also be addressed. How, for example, do Brassica napus monocultures and their intensive pesticide spraying affect the diversity of pollinators in agroecosystems? And how does agrotechnology in other crops affect biodiversity?

Comments on the Quality of English Language

The language is good, but it needs moderate stylistic editing.

Author Response

Comments 1: This article deals with the very important issue of pollination of plant flowers. The review article is of relevance to many plant pollinator researchers. Despite the strengths of the manuscript, I found significant weaknesses and points of discussion that the author should consider.

Response 1: Thanks for the comments

Comments 2: This article deals with the very important issue of pollination of plant flowers. The review article is of relevance to many plant pollinator researchers. Despite the strengths of the manuscript, I found significant weaknesses and points of discussion that the author should consider. 

Response 2: I have eliminated some redundant and unnecessary sentences

Comments 3: Some terms are not used correctly in the text. One such term is 'flora'. For example (line 23) it says: 'in which flora with more abundant rewards'. Flora is a totality of species, an abstract, and therefore cannot be given the characteristics of plants.  

Response 3: Ok, I agree. I changed the generic term flora to plants

Comments 4: The term 'weed' is also used too often, sometimes even when it is not needed. The neutral term 'plant' or 'species' can be used, without negative connotations. Moreover, there is the question of what the author means by 'weed'. 

Response 4: Ok, I agree that the word "weed" is repeated too much. I have replaced many of the terms "weed" with "plant" or "wildflower" in the text so as to avoid considering the presence of wild species as negative.

Comments 5: Can Dianthus carthusianorum really be a 'weed' in the classical sense of the term? I think that the concept of the terms, if the author interprets them freely, should be discussed separately.

Response 5: Ok, I agree that Dianthus cartusianorum it is not a common weed but rather a wild plant from the areas surrounding cultivated fields. Consequently I have changed (caption of figure 5) the term "weed" with "wildflower".

Comments 6: The article is profusely illustrated with original (not credited, but I assume original) photographs. I question whether all the illustrations are really necessary, and whether they are so large (Fig. 4, Fig. 7, Fig. 8, Fig. 9).

Response 6: It is true that I have inserted many photographs (all originals) but I did it with the intention of making the reading more captivating and richer in visual details that can enrich the text with information of immediate understanding

Comments 7: In addition, some of the plants in the photos are misidentified. For example, Fig. 1 (8) is not Malva officinalis (which is no longer the name of the species) but Malva sylvestris; Fig. 1 (11) is not Cirsium vulgare but Cirsium arvense; Fig. 2 (2), not Datura stramonium but Datura innoxia; Fig. 3, the entries are all confused. Other figures also contain misidentified plants. All figures need to be checked and their captions corrected.

Response 7: Yes true... I apologize for the mistakes! I changed all the species that were rightly indicated to me. The only exception I kept Datura stramonium because I have been studying it for many years and I am sure of its identification (also because in the Mediterranean environment Datura innoxia is rather rare). I also corrected the plant species in figure 3.

Comments 8: In addition, some of the plants in the photos are misidentified. For example, Fig. 1 (8) is not Malva officinalis (which is no longer the name of the species) but Malva sylvestris; Fig. 1 (11) is not Cirsium vulgare but Cirsium arvense; Fig. 2 (2), not Datura stramonium but Datura innoxia; Fig. 3, the entries are all confused. Other figures also contain misidentified plants. All figures need to be checked and their captions corrected.

Response 8: It is true that I have inserted many photographs (all originals) but I did it with the intention of making the reading more captivating and richer in visual details that can enrich the text with information of immediate understanding

Comments 9: In addition, some of the plants in the photos are misidentified. For example, Fig. 1 (8) is not Malva officinalis (which is no longer the name of the species) but Malva sylvestris; Fig. 1 (11) is not Cirsium vulgare but Cirsium arvense; Fig. 2 (2), not Datura stramonium but Datura innoxia; Fig. 3, the entries are all confused. Other figures also contain misidentified plants. All figures need to be checked and their captions corrected.

Response 9: Yes true... I apologize for the mistakes! I changed all the species that were rightly indicated to me. The only exception I kept Datura stramonium because I have been studying it for many years and I am sure of its identification (also because in the Mediterranean environment Datura innoxia is rather rare). I also corrected the plant species in figure 3.

Comments 10: I think that the use of vernacular plant names in the text is redundant. They do not provide any information and in some cases only cause confusion. We are well aware that English vernacular names vary widely from one English-speaking country to another, not to mention other countries where English is not the main language. In a scientific article, scientific plant names are quite sufficient.

Response 10: Ok, it's true I had inserted some vulgar terms (for example Common poppy) that I deleted!

Comments 11: Is it not possible to specify the species, or at least the genera, of the insects in the photographs? For example, Fig. 6 has many beetles but no names. 

Response 11: In the last few years I have also included some classifications of pollinating insects but I have understood that the entomological referees do not accept an identification by agronomists considering it presumptuous. Consequently I respect not to invade the field of entomologists and I limit myself to reporting the most easily identifiable taxonomic groups.

Comments 12: Agroecosystems are composed not only of weeds but also of crops. I believe that the impact of crops on the protection of pollinator insect diversity should also be addressed. How, for example, do Brassica napus monocultures and their intensive pesticide spraying affect the diversity of pollinators in agroecosystems? And how does agrotechnology in other crops affect biodiversity?

Response 12: This observation is definitely correct. However, the review, already very long in itself, has the objective of re-evaluating the historical negativity of weeds in terms of "ecosystem service" for pollinators. In other words, I intend to address this important topic (crop rotations, cover crops, pollinated-crops, etc.) in a future review in which these agronomic strategies are also considered, but not derived from "wild flora".

Reviewer 3 Report

Comments and Suggestions for Authors

The manuscript submitted for review concerns the issue of weed role for pollinator in the agroecosystem: plant-insect interactions and agronomic strategies for biodiversity conservation. The topic of biodiversity conservation is very important nowadays. It is also increasingly discussed by various researchers. It is particularly important in the context of the agroecosystem. It is necessary to precisely define the concept of "weeds" in the manuscript. In many research centers it has different meanings. Additionally, the concept of weeds is increasingly being abandoned in favor of the concept of undesirable plants. This is due to the fact that, for example, a rose growing in rye cultivation will be a weed, or rather an undesirable plant. Generally, the manuscript is written correctly. Fig.4 is not very clear, you should add a photo of the (whole) butterfly itself with visible characteristic features.

Author Response

Comments 1: The manuscript submitted for review concerns the issue of weed role for pollinator in the agroecosystem: plant-insect interactions and agronomic strategies for biodiversity conservation. The topic of biodiversity conservation is very important nowadays. It is also increasingly discussed by various researchers. It is particularly important in the context of the agroecosystem. 

Response 1: Thanks for the comments. 

Comments 2: It is necessary to precisely define the concept of "weeds" in the manuscript. In many research centers it has different meanings. Additionally, the concept of weeds is increasingly being abandoned in favor of the concept of undesirable plants. This is due to the fact that, for example, a rose growing in rye cultivation will be a weed, or rather an undesirable plant. Generally, the manuscript is written correctly. 

Comments 3: Fig.4 is not very clear, you should add a photo of the (whole) butterfly itself with visible characteristic features.

Response 3: Ok, I accepted this suggestions about Figure 4. I have not the photo that shows the whole butterfly body. Consequently I inserted the scientific name of this butterfly in the caption (Gonepteryx rhamni). In this case I am certain of the classification of the butterfly as it is very widespread especially where there is Mediterranean scrub as the adult lays eggs on Rhamnus plants.

Round 2

Reviewer 2 Report

Comments and Suggestions for Authors

The article has been substantially revised and most of the comments made in the first review have been addressed. There are still a few editorial points that I would like to bring to the author's attention. Unnecessary capital letters (e.g. line 117) should be corrected.

1. Frequent technical errors should be corrected (missing spaces between words, incorrect fonts (names of authors of taxa unnecessarily in italics), unnecessary or missing dots. Also, captions for figures should use a dash rather than an equal sign, and individual elements should be separated by commas or semicolons.

2. Since the author often uses taxonomy and plant names different from the current taxonomy, the source of the scientific names of plants should be indicated.

3.  I recommend that the author of the figures be indicated at the end of the caption (even if there is repetition, the figures should stand alone and contain all necessary information).

Comments on the Quality of English Language

The English is generally good, but the occasional awkward sentences should be edited. This would only improve the value and clarity of the article.

Author Response

The article has been substantially revised and most of the comments made in the first review have been addressed. There are still a few editorial points that I would like to bring to the author's attention. Unnecessary capital letters (e.g. line 117) should be corrected.

  1. Frequent technical errors should be corrected (missing spaces between words, incorrect fonts (names of authors of taxa unnecessarily in italics), unnecessary or missing dots. Also, captions for figures should use a dash rather than an equal sign, and individual elements should be separated by commas or semicolons.

Response: Ok, I have corrected all suggestion indicated in the pdf version of the paper.

  1. Since the author often uses taxonomy and plant names different from the current taxonomy, the source of the scientific names of plants should be indicated.

Response: Ok, in both tables I indicated the the source of the scientific names.

  1. I recommend that the author of the figures be indicated at the end of the caption (even if there is repetition, the figures should stand alone and contain all necessary information).

Response: Ok, the author of the photos was reported in all figures.